# Transcriptome Analysis Reveals Candidate Genes Involved in Light-Induced Primordium Differentiation in *Pleurotus eryngii*

**DOI:** 10.3390/ijms23010435

**Published:** 2021-12-31

**Authors:** Dou Ye, Fang Du, Qingxiu Hu, Yajie Zou, Xue Bai

**Affiliations:** 1Institute of Agricultural Resources and Regional Planning, Chinese Academy of Agricultural Sciences, Beijing 100081, China; yedou363216@126.com (D.Y.); duf070413@126.com (F.D.); zouyajie@caas.cn (Y.Z.); 2College of Life Science and Technology, Huazhong Agricultural University, Wuhan 430070, China; 3China National Center for Bioinformation/Beijing Institute of Genomics, Chinese Academy of Sciences, Beijing 100101, China; 4College of Forestry, Central South University of Forestry and Technology, Changsha 410004, China; Baxulucky@163.com

**Keywords:** high-throughput sequencing, *Pleurotus eryngii*, primordium differentiation, *WC-1*, PHR

## Abstract

*Pleurotus eryngii*, a highly valued edible fungus, is one of the major commercially cultivated mushrooms in China. The development of *P. eryngii*, especially during the stage of primordium differentiation, is easily affected by light. However, the molecular mechanism underlying the response of primordium differentiation to light remains unknown. In the present study, primordium expression profiles under blue-light stimulation, red-light stimulation, and exposure to darkness were compared using high-throughput sequencing. A total of 16,321 differentially expressed genes (DEGs) were identified from three comparisons. GO enrichment analysis showed that a large number of DEGs were related to light stimulation and amino acid biosynthesis. KEGG analyses demonstrated that the MAPK signaling pathway, oxidative phosphorylation pathway, and RNA transport were most active during primordium differentiation. Furthermore, it was predicted that the blue-light photoreceptor *WC-1* and Deoxyribodipyrimidine photolyase *PHR* play important roles in the primordium differentiation of *P. eryngii*. Taken together, the results of this study provide a speculative mechanism that light induces primordium differentiation and a foundation for further research on fruiting body development in *P. eryngii*.

## 1. Introduction

*P. eryngii*, also known as the king oyster mushroom, is one of the major commercially cultivated mushrooms in China [1]. *P. eryngii* is rich in proteins and essential amino acids, and highly appreciated by consumers for its unique almond aroma and abalone flavor [2].

Previous studies have demonstrated that primordium formation and differentiation in *P. eryngii* can be markedly affected by environmental factors, especially light. Specifically, *P. eryngii* primordium could not differentiate from a completely dark environment [3]. Blue light stimulates the differentiation of the primordium, while red light exerts an inhibitory effect [4,5]. Nevertheless, the mechanism underlying the response of *P. eryngii* to light during primordium differentiation has not been elucidated. In recent years, a breakthrough was made in research on the light reaction mechanism of *Cordyceps militaris*. Wang et al. found that in *C. militaris*, photoreceptor knockout mutants failed to form primordia under light exposure, showing that the photo-response mechanism of *C. militaris* primordium formation directly involves photoreceptors [6]. A comparative transcriptomics analysis of *Pleurotus ostreatus* revealed that blue light induces primordium differentiation by enhancing the activation of glycolysis and the pentose phosphate pathway, whereas red light weakens glycolysis and pentose phosphate pathway activation to impede this stage of development [7]. Blue light plays a dominant role in *Lentinula edodes* and required for the transformation of nutritional mycelia into a brown film [8]. The depth and thickness of this brown film directly affect the development of the primordium [9].

In recent years, genome-wide sequencing provides an opportunity to explore organism development mechanism, which has been widely used in *Capsicum annuum* [10], *Triticum aestivum* L. [10,11], soybean [12], and *Salvia miltiorrhiz*a [13]. Unlike genome-wide sequencing, RNA-sequencing (RNA-seq) is a superior technology to explore functional genomic data for organisms due to transcript can present specific gene expression, so transcriptome sequencing can be performed as a straightforward method for mining differential transcript properties of organisms under different conditions [14]. RNA-seq was used to investigate the tolerance properties of *P. eryngii* to CdCl_2_ and NO, and concluded that genes related to oxidoreductase, dehydrogenase, reductase, transferase, and transcription factors can contribute to enhancing the tolerance of *P. eryngii* to heavy metals [15]. A comparative transcriptome analysis of mature and immature *Pleurotus tuoliensis* mycelia showed that nucleotide synthesis and energy metabolism were highly active during mycelial maturation and the gene nucleoside diphosphate kinase (*NDPK*) performed as a key role in this physiological process [16]. In *Ganoderma lucidum*, a comparative transcriptome analysis of normal mycelia and mycelia subjected to heat stress for 2 h revealed that genes associated with stress resistance, protein assembly, transport and degradation, signal transduction, carbohydrate metabolism and the energy supply were all upregulated in mycelia after heat stress, and this research laid the foundation for the mining of heat stress response genes in *G. lucidum* [17]. In a study of dikaryotic mycelia and mature fruiting bodies in *L. edodes*, RNA-seq analysis showed that fruiting body-specific transcripts were significantly enriched in replication, repair, and transcription pathways, which are important for premeiotic replication, nuclear division and meiosis during maturation [18]. To reveal the regulatory mechanism of fruiting body development in *P. tuoliensis*, high-throughput sequencing was performed on mycelia, primordia, and fruiting bodies, and the most pronounced change in gene expression occurred during the vegetative-to-reproductive transition (from primordia to fruiting bodies), suggesting that primordium differentiation is the most active stage of development [19].

In this study, Illumina sequencing technology were used to explore the differential properties of *P. eryngii* primordia under three different conditions involving exposure to red light, blue light, and darkness, with the aim of establishing an important basis for elucidating the mechanism of primordium differentiation in *P. eryngii.*

## 2. Results

### 2.1. Morphological Features of Primordia under Different Light Conditions

Cultivation bags fully covered by mature mycelium were cultured in the CNC cultivation house. As shown in Figure 1, the mycelia started to twist and form primordia after approximately 9 days under red light, blue light, and dark conditions. However, significant differences among the different light conditions were observed in the subsequent differentiation stage. Under dark conditions, the primordia exhibited a rough surface and had difficulty forming normal buds, producing malformed young buds without obvious pilei and stalks. In contrast, the primordia under red light and blue light conditions differentiated normally, and young buds were produced in greater abundance and faster under blue light than under red light.

### 2.2. Sequencing

The raw Illumina sequencing data were deposited in GenBank under BioProject accession PRJNA759008. Table 1 provides an overview of the transcript-level data. After excluding low-quality reads, a total of 21.96 GB of clean data were obtained. High-quality reads showed a mapping rate of 91%, which was above the standard mapping rate of 70%. Moreover, the Q30 values of all sequences of the nine cDNA libraries exceeded 90%, and more than 90% of the bases were valid. Boxplots (Figure 2) showed that the three samples in each group yielded a concentrated range of values and that the sample similarity was high. The above results indicated that the transcripts of the nine primordium samples were all with high quality and could be used for subsequent analysis.

### 2.3. DEG Identification and Functional Annotation

To investigate transcript differences during primordium differentiation among the different light conditions, DEGs among the groups were identified and annotated. A total of 16,321 DEGs were identified among the three conditions (Figure 3). Specifically, 2797 upregulated and 3608 downregulated in R vs. B, 2810 upregulated and 2545 downregulated in D vs. R, 2044 upregulated and 2517 downregulated in D vs. B. To determine the functions of these DEGs, GO enrichment analysis and KEGG functional annotation were performed, and the results are shown in Figure 4.

In the D vs. B comparison, the 4561 DEGs were enriched in 3911 terms, including 587 terms in the cellular component category, 1265 terms in the molecular function category and 2059 terms in the biological process category. In the D vs. R comparison, 5355 DEGs were enriched in 4401 terms, including 675 terms in the cellular component category, 1331 terms in the molecular function category and 2395 terms in the biological process category. These results are displayed in Appendix A. There were 16 GO terms related to light, which are presented in Figure 5 and Appendix A. Among these GO terms, deoxyribodipyrimidine photolyase activity (GO: 0003904), response to light stimulus (GO: 0009416), and photoreceptor connecting cilium (GO: 0032391) were significantly enriched in both the D vs. B and D vs. R comparisons, implying that light has a great influence on primordium differentiation and that this influence may be exerted through deoxyribodipyrimidine photolyase. The KEGG pathway enrichment results for the D vs. B comparison and the D vs. R comparison are provided in Appendix A, respectively. We focused on the top 20 pathways (Figure 6) and found that the DEGs in these two comparisons were highly concentrated in the KEGG pathways MAPK signaling pathway (MSPY; ko04011), oxidative phosphorylation (OP; ko00190), and RNA transport (RT; ko03013), implying that signal transduction, material and energy metabolism remained significantly active during primordium differentiation in *P. eryngii*; this finding deserves special attention.

In the R vs. B comparison, 6406 DEGs were enriched in 4754 terms, including 697 terms in the cellular component category, 1441 terms in the molecular function category, and 2616 terms in the biological process category; the results are shown in Appendix A. Figure 7A shows the top 30 GO enrichment terms of upregulated genes, which include 11 terms in the biological process category, 2 in the cellular component category, and 17 in the molecular function category (*p* < 0.05). Figure 7B shows the top 30 GO terms of downregulated genes, including 17 terms in the biological process category, 4 terms in the cellular component category, and 9 terms in the molecular function category. The terms serine family amino acid biosynthetic processes (GO: 0009070), L-serine biosynthetic processes (GO: 0006564), the positive regulation of phosphatidylcholine biosynthetic processes (GO: 2001247), acetyl-CoA biosynthetic processes (GO: 0006085), and ADP biosynthetic processes (GO: 0006172) showed significant enrichment during blue light induced primordium formation. Given the strong promotion of primordium differentiation under blue light, we conjecture that the positive effect of blue light illumination may be exerted through these biosynthetic pathways.

The results of the KEGG pathway enrichment analysis are displayed in Appendix A. The DEGs upregulated under blue light were enriched in 177 KEGG pathways, and the DEGs downregulated under blue light were enriched in 187 KEGG pathways. Figure 8 illustrates the top 20 enriched KEGG pathways. The DEGs were again found to be significantly enriched in the KEGG pathways RT, MSPY, and OP. In the blue-light primordium, the numbers of upregulated DEGs enriched in the RT, MSPY, and OP pathways were 63, 49 and 33, respectively. In the red-light primordium, 43 DEGs in the RT pathway were upregulated, 51 DEGs in the pathway MSPY were upregulated, and 38 DEGs in the OP pathway were upregulated. In some cases, both upregulated genes and downregulated genes were enriched in the same pathway, and cultivation under blue light was shown to promote primordium differentiation, while red light had the opposite effect. Thus, we speculate that these pathways are related to the physiological process of primordium differentiation in *P. eryngii*.

### 2.4. Analysis of DEGs Involved in Light-Induced Primordium Differentiation

To explore the mechanism of primordium differentiation under scattered-light stimulation, the red-light primordium transcripts and blue-light primordium transcripts were separately compared with the dark primordium transcripts. Through comparative analysis, two groups of DEGs were obtained, which are presented in Figure 9.

First, the DEGs associated with light were analyzed. In the D vs. R comparison, we noted that the gene encoding quinone oxidoreductase *ZTA1*, which related to high light intensity (GO: 0009644), was significantly upregulated in the primordium exposed to red light. Another interesting result was that in the D vs. B comparison, the *PHR* (878624) gene, encoding deoxyribodipyrimidine photolyase, was significantly upregulated 33-fold in the blue-light primordium.

Immediately thereafter, we analyzed MAPK signaling pathway-related genes. We found that the expression of the gene *STEA* was upregulated under the light conditions in both comparisons, suggesting that this transcription factor is active only under light. Catalase-1 (CAT-*1*) detoxifies H_2_O_2_ by transforming it into water and oxygen, which can promote the oxidative stress response of fungi to light stimulation [18,20]. As expected, the gene expression of *CAT-1* was upregulated under light in both comparisons. Then, the genes enriched in the oxidative phosphorylation pathway were analyzed. A class of genes encoding V-type proton ATPases showed upregulation in all groups, indicating that these enzymes have little relationship to light stimulation and are primarily involved in cell construction. Succinate dehydrogenase is an important enzyme positioned between oxidative phosphorylation and electron transfer and was upregulated under light in both the D vs. B comparison and the D vs. R comparison. We speculate that the high expression level of succinate dehydrogenase was caused by an increase in the oxidative stress response due to light stimulation.

Finally, numerous genes enriched in RNA transport pathways were found, such as encoding an ATP-dependent RNA helicase TIF1, encoding a translation initiation factor *TIF224*, and encoding eukaryotic translation initiation factor 4E-1 TIF451. Gene expression involves two main steps: transcription and translation. The results imply that primordium differentiation involves the expression of many proteins.

### 2.5. Analysis of DEGs during Primordium Differentiation Related to Red and Blue Light

As the primordium differentiation rate and quantity were significantly higher under blue-light illumination than under red-light illumination, a comparative analysis of red-light primordium transcripts and blue-light primordium transcripts was performed to investigate the key genes involved. We focused on 13 genes related to light stimulation: 9 genes (*WC-1*: 1164413, *PHR*: 1391851, *PHR*: 1391738, *PHR*: 1447773, *ZTA1*: 1350178, *PHR*: 15069953, *TFCC*: 1432018, *TFCC:* 1483280, *TFCC*: 1507938) upregulated in blue-light primordium samples and 4 genes (*AGO1*: 1445782, *OGG1*: 1486735, *SR45A*: 1557381, *ZTA1*: 1234511) upregulated in red-light primordium samples (Table 2). GO enrichment analysis showed that *WC-1* was related to photoreceptor activity (GO: 0009881), transcription factor activity (GO: 0003700) and protein-chromophore linkage (GO: 0018298) and *TFCC* was associated with the cellular component term photoreceptor connecting cilium (GO: 0032391). *PHR* was upregulated by nearly 136-fold in blue-light primordium and was enriched in both the biological process terms deoxyribodipyrimidine photolyase activity (GO: 0003904) and protein-chromophore linkage (GO: 0018298).

Considering the specificity of the three KEGG pathways (RT, MSPY, OP), candidate genes in these pathways were investigated and presented in Appendix A. In the RT pathway, the expression of *UPF1* (1377433), and *TIF221* (1366755), and some genes encoding nuclear proteins *NUP192* (1398997), *NUP45* (1423814), *NUP192* (1475344), *NUP85* (1480758), *NUP132* (1501787), and *NUP189* (1385072) were upregulated in blue-light primordium, while these genes encoding nuclear proteins were expressed at lower levels in the red-light treatment than in the blue-light treatment, suggesting that the relative inhibition of primordium differentiation by red light may result from the blockade of nuclear protein synthesis. In the pathway MSPY, the expression levels of the transcription factor *STEA*, developmental regulator *FLBA*, and several kinases, including *SKH1* (1374469), *CDC25* (1382011), *WIS1* (1378155), *MKK1* (1416976), *CKI1* (1367085), and *STT4* (580687), were upregulated in the blue-light primordium. In the OP pathway, the gene encoding V-type proton ATPase *VMA5* (75310) was significantly upregulated in blue-light primordia, while the expression levels of other V-type proton ATPase genes (*VMA1* 1390441, *VMA3* 1456048, *VMA13* 1433531) were upregulated in red-light primordia.

### 2.6. Protein-Protein Interaction (PPI) Network Prediction

The STRING database of known and predicted protein interactions were used to find potential relationships between proteins in primordium differentiation analysis. Proteins (*WC-1*, *PHR*, *ZTA1*, *TFCC*, *AGO1*, *OGG1*, *SR45A*) that may respond to light analyzed in this research were used as input to STRING. Selected proteins were linked by known and predicted interactions, and a small network was established (Figure 10). In this network, those known proteins and other unknown interaction proteins were showed. Blue light photoreceptors *WC-1* and *PHR* which we focused were predicted as neighborhood gene. At the same times, we found that proteins RNF41, NTH1, A0A369K0D4 interacted with *PHR*, and the protein CGPB interacted with *WC-1*.

### 2.7. Real-Time PCR Validation of Transcription Data

To validate the RNA-seq data, we performed quantitative RT-qPCR with the same primordium tissue. We randomly selected a dozen genes to verify the relative expression levels, including some genes that were upregulated under light and some that were downregulated. Figure 11 shows that the qPCR data were very consistent with the transcriptome sequencing data in the three comparisons, suggesting that the transcriptome data were reliable.

## 3. Discussion and Conclusions

*P**. eryngii* is a typical wood rot fungus that mainly relies on mycelia to secrete extracellular enzymes (laccase, etc.) to degrade lignocellulose in the substrate to maintain growth, to some degree, the degradation are related to the morphogenesis of the fruiting-body [21,22]. In the present study, we concentrated on primordium differentiation induce by light, *P. eryngii* were cultivated with identical substrates to avoid degradation interference.

Primordium formation and differentiation are the critical stages before the growth and development of the fruiting body and are extremely sensitive to light. As early as 1968, the scientist Newman reported that dark conditions inhibited the mycelial growth of *Melanotus* sp. and the mycelia could only twist to form the primordium and fruiting body under light incubation [23]. The rare edible fungus *Isaria cicadae* was shown to have differential responses to different light conditions under cultivation: neither primordia nor synnemata formed under red-light irradiation, and primordia formed only under blue light [24]. Our studies demonstrated that the *P. eryngii* primordium could form under dark conditions but was largely unable to subsequently differentiate into young buds with a stalk and cap structure. The differentiation of *P. eryngii* primordium depended on light stimulation and occurred earlier under blue-light induction than under red-light induction; this finding is contrast to the research on *Cordyceps militaris* which illustrated red-light exposure could shorten the time of primordium formation [25]. and we speculate that this difference is due to species differences in the response to light wavelengths. Primordium formation and differentiation in *Coprinus stercorarius* required blue-light, but continuous light exposure inhibited these processes; only alternating periods of light and darkness could exert a development-promoting effect [26]. Periodic light exposure was also applied in this study, as we took into account the periodic and extremely regular light conditions observed in nature. The yield of mushrooms and the development of the stalk and cap are directly influenced by light conditions. The yield of *P. ostreatus* is higher under blue light than under red light due to the superior development of stalks and caps under blue light [7].

Complex systems perceive and transform red- or blue-light signals in fungi; nevertheless, the molecular mechanisms by which light activates or inhibits primordium differentiation remain poorly understood. In this study, several important biosynthetic pathways, including the amino acid biosynthetic pathway, the positive regulation of the phosphatidylcholine biosynthesis pathway, and the acetyl Co-A biosynthetic pathway, were found to be active in the blue-light primordium samples, as expected, relatively inactive in the red-light primordium samples. Amino acids are used for the synthesis of tissue proteins and hormones and serve as energy-producing substances in organisms [27]. Phosphatidylcholine is an significant component of cell membranes and plays an important role in cell signaling pathways [28,29]. Acetyl-CoA plays an essential role in glycolysis and fatty acid oxidation [30]. The significant enrichment of these biological pathways suggests that they probably play active roles in blue-light-induced mycelial twisting and primordium formation. However, the mechanisms by which light is sensed and regulates biosynthesis need to be revealed by in-depth research.

Previous studies have shown that many KEGG pathways are related to the growth and development of fungi, such as the MAPK [31,32], oxidative phosphorylation [33], RNA transport and cAMP-PKA pathways [34]. In *P**. eryngii subsp. tuoliensis* (Bailinggu), cold stimulation is the main factor for triggering primordia initiation, and MAPK pathways play a vital role in response to cold stimulation [35]. According to the KEGG enrichment analysis of the DEGs induced by red light and blue light, multiple nucleoprotein-encoding genes were upregulated in the blue-light primordium samples, confirming that nucleoproteins are required for the formation of the nucleus during the unique developmental process of primordium formation. Various kinases involved in the MAPK signaling pathway were also upregulated in the blue-light primordium samples. In most fungi, including *P. ostreatus* [7] and *Cordyceps militaris* [36], blue light can induce and accelerate primordium formation, and the related kinases in the MAPK signaling pathway were significantly upregulated in this stage. Another gene enriched in the MAPK signaling pathway, FLBA, which is a developmental regulator to activate the FadA GTPase [37], also showed high expression in the blue-light primordium samples, showing that the activation of the MAPK signaling pathway plays an important role in the rapid formation of primordium. The expression of *STEA* was upregulated in red- and blue-light primordium samples compared with dark primordium samples. Perhaps the most thorough account of *STEA* is from research on *Arthroderma benhamiae*. Krober employed an RNA interference technique to knock down *STEA* expression, and *A. benhamia* with loss of *STEA* did not produce hair perforation organs, but mycelial formation during growth on hair and nails was unaffected [38]. Previous studies suggest that the *STEA* gene might be involved in the unique development of sex organs in *Aspergillus niger* and *Aspergillus nidulans*, but its association with mycelium growth has not been reported [39,40]. In the present research, we speculated that the primordium was only able to differentiate under illumination due to the high expression of *STEA* under light. In contrast, although primordium formed easily in the absence of light, differentiation was inhibited, possibly due to the low expression of *STEA*. The expression of one category of *VMA* genes encoding V-type proton ATPases in primordia was upregulated under both light conditions relative to the dark condition. V-type proton ATPases act specifically as ATP-driven proton pumps, which are essential for the normal development of organisms [41].

*ZTA1* plays a key role in the regulation of oxidative stress in fungi. In yeast, disruption of the *ZTA1* gene has been found to have no effect on cell growth under standard conditions but to make yeast more sensitive to oxidative stress agents [42]. In this study, *ZTA1* was highly expressed in red-light primordium, indicating that the primordium cells were exposed to hyperoxidative stress upon red-light irradiation. Transcription factors that regulate the blue-light response have been identified and studied deeply in fungi and plants, such as *WC-1* and *PHR*. The literature has highlighted that *WC-1* is a blue-light photoreceptor with transcription factor activity, while *PHR* is a blue-light photoreceptor with photolytic enzyme activity that may be involved in sensing blue wavelengths to repair DNA damage [43]. The mechanism by which *WC-1* and *PHR* in edible fungi senses light stimulation has been a hot research topic in recent years. The blue-light receptor gene *CmWC-1* in *C. militaris* has been suggested to promote the switch from a nutritional growth state to primordium formation by inhibiting steroid biosynthesis [44]. External blue-light stimulation can increase the expression of *Glwc-1* and *Glwc-2* during *G. lucidum* primordium differentiation, which suggests that the blue-light photoreceptors *WC-1* and WC-2 exert essential functions in primordium differentiation [45]. *PoWC-1* and *PoWC-2* in *P. ostreatus* have been confirmed to regulate the expression of *P. ostreatus* target genes and control its development; to exert their biological functions, the two photoreceptors must bind flavin adenine dinucleotide (FAD) to form a complex [46]. In *Trichoderma atroviride*, blue-light photoreceptors BLR-2 and BLR-1 localize to the *PHR*-1 promoter and can function as transcriptional complexes in response to light [47]. *Neurospora crassa* has been widely used as a model organism for light response study, which contain two blue light photoreceptor genes *WC-1* and *WC-2* [48,49]. The formation of *N. crassa* conidia requires the genes *CON-10* and *CON-6*, and the activation of this gene requires the transcription complex (WCC) formed by photoreceptors *WC-1* and *WC-2*, only under light, the WCC complex binds to the gene promoter to activate transcription [50]. Another typical ascomycetes *T**uber borchii*, whose apical growth was inhibited under blue-light irradiation, contains one blue light photoreceptor *WC-1* with high similarity to *NcWC-1* [51] In our research, As show in Figure 12, *WC-1* and *PHR* showed high expression levels in blue-light primordia when compared to red-light primordium and darkness primordium, speculating that the blue-light receptor *WC-1* may be an important transcription factor involved in the differentiation of blue-light-induced primordia and that *PHR* may be an important enzyme involved in the primordium differentiation induced by blue light. 

Our previous research on pileus morphogenesis analysis of *P. eryngii* under different lights also showed that *PHR* exhibited high expression level in pileus exposed to blue light, while the expression level of *WC-1* had no significant difference under different lights. Primordium differentiation and pileus morphogenesis are two different developmental stages of *P. eryngii*, but both require light stimulation, so genes involved in sensing and transducing light were significantly upregulated in these two stages, along with developmental properties-related genes, such as pileus-specific protein hydrophobin-encoding gene, tyrosinase-encoding genes in pileus, quinone oxidoreductase encoding genes and developmental regulator encoding genes in primordium, all showing high expression levels. Accordingly, pathways involved in light response and signal transduction in these two stages were all active, like MAPK signal pathway [52]. The growth and development of all living organisms are complex and dynamic. Therefore, whether there was an interaction between *WC-1* and *PHR* in *P. eryngii* needs to be further investigated.

In summary, we obtained 16,321 DEGs through a transcriptome sequencing analysis of three categories of primordia. GO and KEGG enrichment analyses showed that the DEGs identified in response to light treatment were mainly enriched in pathways associated with amino acid biosynthesis. Additionally, genes encoding nucleoproteins were significantly upregulated in blue-light primordium samples compared with red-light primordium samples. We speculate that these DEGs may be related to the rate of cell construction. Furthermore, we observed that genes associated with photoresponse characteristics and that encode white-collar protein and deoxyribodipyrimidine photolytic enzyme, were all significantly upregulated in blue-light primordium samples. The results of this study lay a solid foundation for photophysiological research on the *P. eryngii* primordium and suggest directions for the further study of fruiting body development in *P. eryngii.*

## 4. Materials and Methods

### 4.1. Culture Conditions and Primordium Collection

The commercial *P. eryngii* strain ACCC_52611 was provided by the Agricultural Culture Collection of China. The preserved strain was activated on potato dextrose agar (PDA) medium before the cultivation experiments. *P. eryngii* mycelia were grown at 25 °C in polypropylene cultivation bags (size: 42 × 22 cm) containing 1350 g of cultivation substrate (26.8% cotton seed hull, 26.8% sawdust, 11.1% corn flour, 11.1% soybean meal, 22.2% wheat bran, 1% calcium carbonate, 1% lime, 65% water content, pH 8.0–9.0) in the dark for 28 days, during which the substrates were fully covered by mycelia. Then, the cultivation bags were distributed evenly and randomly in three mushroom cultivation houses with computer numerical control (CNC) for subsequent fruiting management. The light conditions of the three cultivation houses were set to 12 h red-light/dark periodic illumination, 12 h blue-light/dark periodic illumination, and 24 h dark conditions. Temperature and humidity were controlled at 12–14 °C and 90%, respectively, until primordia formed. Three cultivation bags were randomly selected from each cultivation house, and primordium samples were collected with a sterilized knife, placed in sterile Eppendorf (EP) tubes, immediately frozen in liquid nitrogen and then preserved frozen at −80 °C for RNA extraction. For the convenience of reporting, the primordia from the red-light house, blue-light house, and dark house were referred to as R, B, and D, respectively.

### 4.2. RNA Extraction, cDNA Library Construction and Illumina Sequencing

Frozen samples were fully ground in liquid nitrogen, and total RNA was extracted according to the protocol of the mirVanaTM miRNA Isolation Kit (Ambion, Elk Grove, CA, USA). After the remaining gDNA in the RNA was digested with DNase, mRNA was enriched with oligo (dT) magnetic beads, and fragmentation buffer was then added to fragment the mRNA into short pieces. First-strand cDNA synthesis was conducted with random hexamer primers using the short mRNA fragments as templates, and second-strand cDNA synthesis was performed using the first-strand cDNA as a template. The purified double-stranded cDNA ends were repaired, a-tailed and ligated with sequencing adapters. Then, fragment size selection was performed to construct a library through PCR amplification. The nine cDNA libraries were assessed for quality on an Agilent 2100 Bioanalyzer (Agilent Technologies, Santa Clara, CA, USA) and sequenced using an Illumina HiSeqTM 2500 system (HiSeq^TM^ 2500, San Diego, CA, USA)).

### 4.3. Differentially Expressed Gene (DEG) Analysis and Functional Annotation

The raw reads generated via high-throughput sequencing were provided as fastq format sequences. The raw reads were filtered by adapter removal, and low-quality bases and N bases were removed by using NGS QC Toolkit software [53] to obtain high-quality clean reads. Subsequently, the clean reads were mapped to the reference genome of *P. eryngii* (https://genome.jgi.doe.gov/portal/Pleery1/download/Pleery1_AssemblyScaffolds.fasta.gz (accessed on 6 September 2018) using HISAT2 (http://ccb.jhu.edu/software/hisat2 (accessed on 6 September 2018)) with the default software parameters, and the samples were evaluated according to the total mapping ratio.

The present study was carried out at the transcriptome level. Sequence similarity searches were performed, and clean reads were mapped to the *P. eryngii* transcripts derived from the JGI transcriptome reference database (https://genome.jgi.doe.gov/Pleery1/download/Pleery1_all_transcripts_20150629.nt.fasta.gz (accessed on 6 September 2018) and annotated with the GFF2 annotation file (https://genome.jgi.doe.gov/Pleery1/download/Pleery1_all_genes_20150629.gff.gz (accessed on 6 September 2018). Transcript expression levels were calculated as fragments per kb per million reads (FPKM) values using eXpress, and the counts of protein-coding reads were analysed using Bowtie2. Differentially expressed transcripts were identified using the estimate size factors function of the DESeq package of R. *P*-values and fold changes between samples were computed using the nbinom test function. Differences were considered significant according to a *p*-value < 0.05 and a |log2−fold change| > 1.

Enrichment analyses were performed based on Gene Ontology (GO) [54] and Kyoto Encyclopedia of Genes and Genomes (KEGG) annotation [55]. The number of differentially expressed transcripts allocated to each GO term was counted, and the significance of the enrichment of differentially expressed transcripts under each GO term was calculated using the hypergeometric distribution test. The test result returned a *p*-value, with a small *p*-value indicating significant enrichment of differentially expressed transcripts under that GO term. At the same time, multiple hypothesis testing correction with *p*-values was performed using the false discovery rate (FDR). KEGG (https://www.genome.jp/kegg/, accessed on 6 September 2018) is a major public pathway-related database. The significance of KEGG pathways was analysed using the hypergeometric distribution test, and the *p*-values were corrected using the FDR.

### 4.4. Real-Time Quantitative (RT-qPCR) Analysis

The accuracy of the RNA-seq data was confirmed by RT-qPCR. Total RNA from primordium samples was extracted using an RNA reagent kit (Omega, GA, USA). Each RNA sample was subjected to RNase-free DNase I (Omega, GA, USA) digestion to remove gDNA, and cDNA was synthesized according to the protocol of the PrimeScript RT Reagent Kit (Vazyme, Nanjing, China). RT-qPCR primers were designed with Primer 5 and are shown in Table 3. The *GAPDH* gene was used as the internal control. Each experiment conducted in this research was repeated three times to verify the reproducibility of the results.

### 4.5. PPI Network

The PPI network of DEGs related to light were constructed using STRING database (http://string-db.org (accessed on 25 November 2021). Gene symbols were used as input and fungi were selected as species. At the same time, the interaction score set as 0.400.

### 4.6. Statistical Analysis

Relative gene expression levels were calculated by the 2^−ΔΔCt^ method. Data were analysed using Excel 2010. Figures were produced using GraphPad Prism software and Adobe Illustrator software. Differences with a *p*-value less than 0.05 were considered statistically significant.

## Figures and Tables

**Figure 1 ijms-23-00435-f001:**
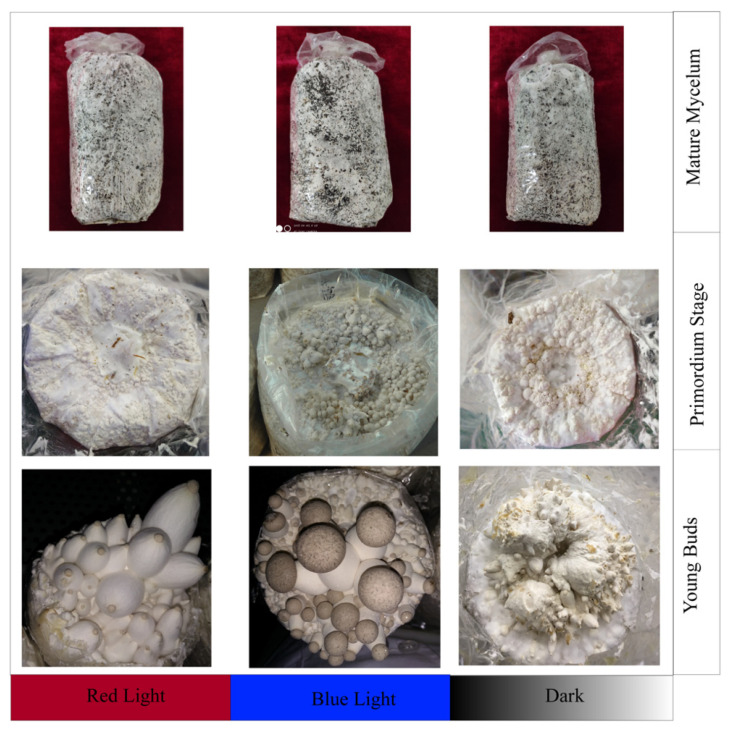
Morphological features of *P. eryngii* at each stage under different light treatments. The different light treatments used in the experiment, red light, blue light, and darkness, are presented along the abscissa. The different development periods are presented along the ordinate.

**Figure 2 ijms-23-00435-f002:**
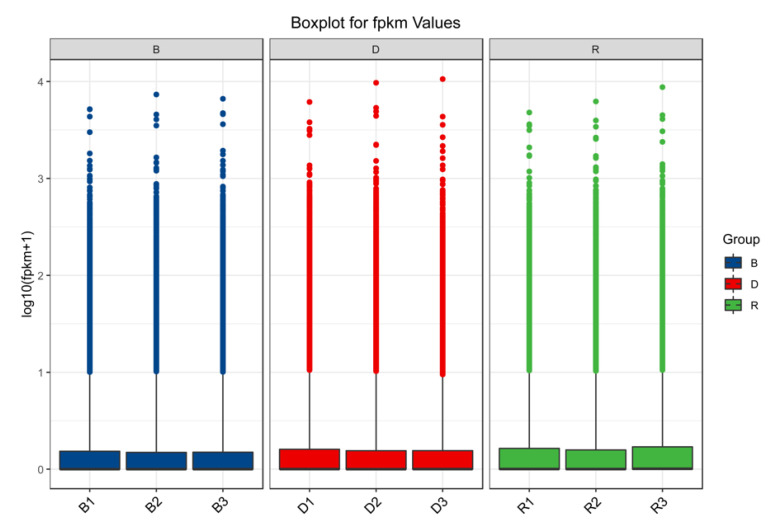
Transcriptional relationships among 9 samples.

**Figure 3 ijms-23-00435-f003:**
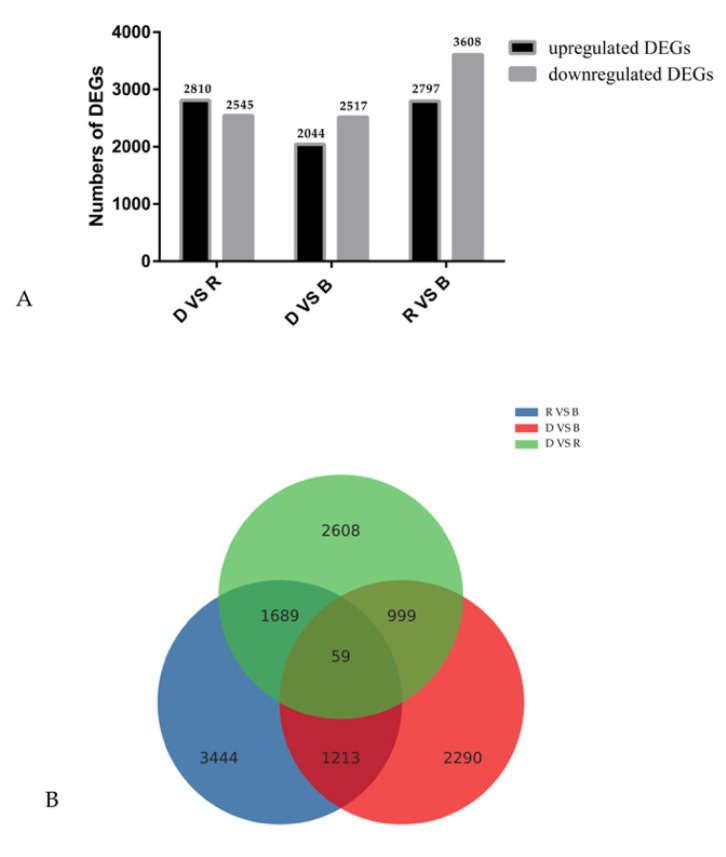
Gene expression comparisons. (**A**). The gene expression profile of DEGs. (**B**). Venn diagram of the number of differentially expressed genes (DEGs). The numbers of upregulated and downregulated genes between D vs. B, D vs. R, and R vs. B are shown.

**Figure 4 ijms-23-00435-f004:**
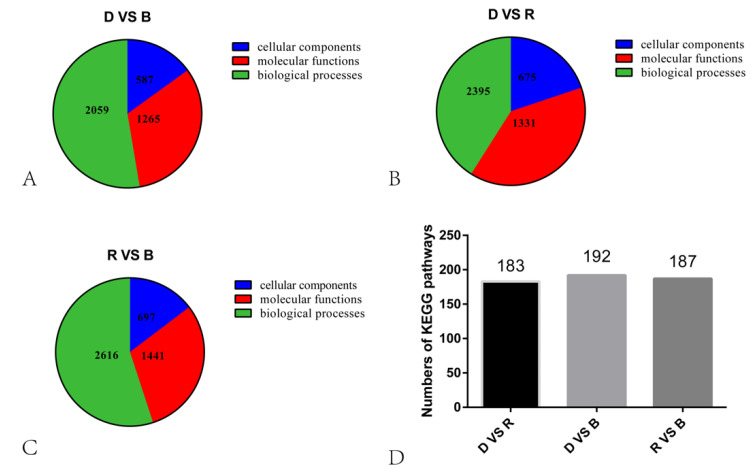
Statistical results of KEGG and GO analyses. (**A**–**C**). Pie chart showing the numbers of GO terms across the three categories. (**D**). The bars indicate the numbers of KEGG pathways.

**Figure 5 ijms-23-00435-f005:**
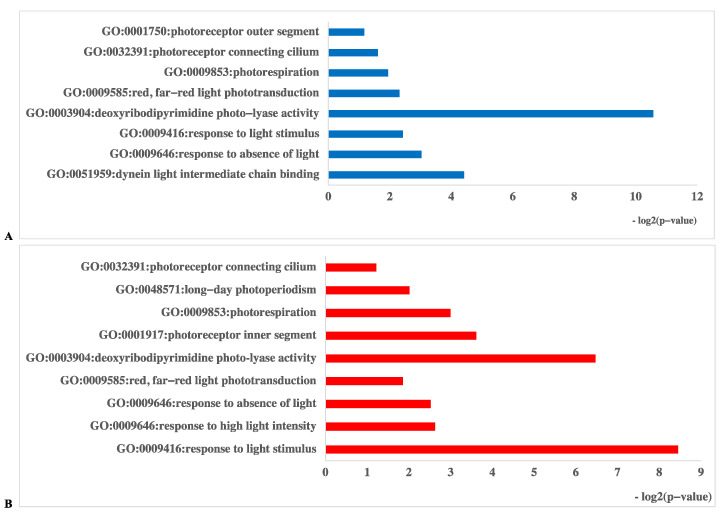
Gene Ontology (GO) database enrichment analysis of DEGs in primordium between blue or red-light conditions and darkness. (**A**). DEGs between D and B. (**B**). DEGs between D and R.

**Figure 6 ijms-23-00435-f006:**
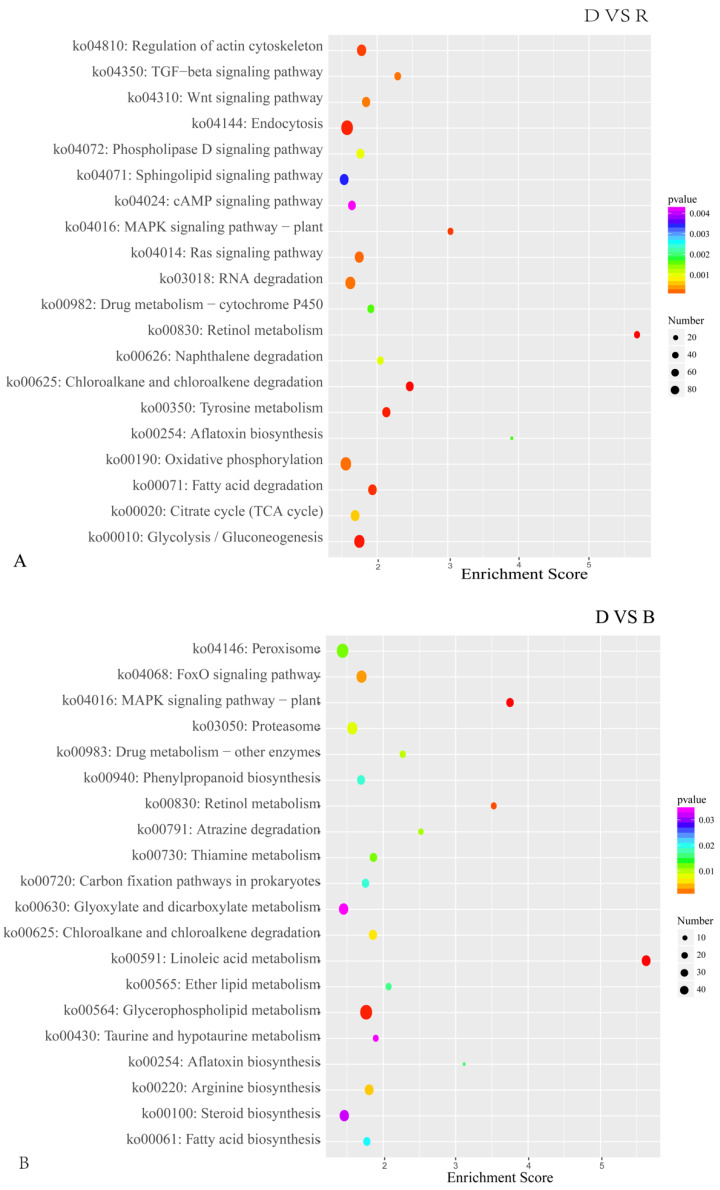
The top 20 enriched KEGG pathways of DEGs in primordium between blue or red light and darkness. (**A**). DEGs between D and R. (**B**). DEGs between D and B. The color intensity is proportional to the enrichment significance, and the circle size indicates the number of enriched genes.

**Figure 7 ijms-23-00435-f007:**
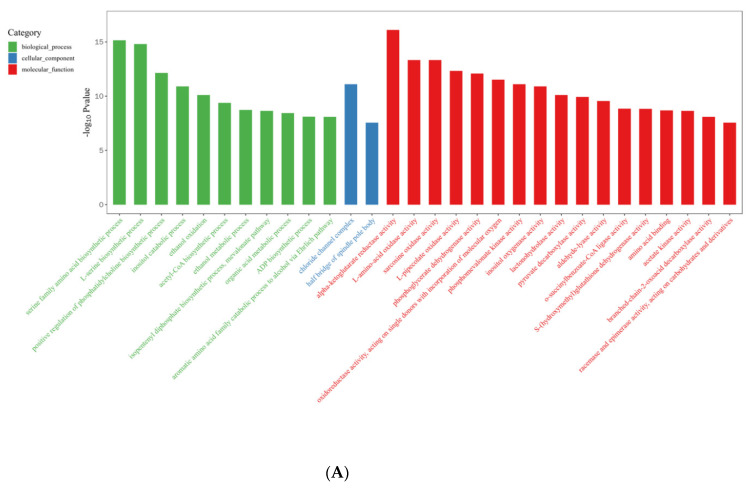
Gene Ontology (GO) database enrichment analysis of DEGs in primordium between blue light and red light. (**A**). DEGs upregulated under blue light. (**B**) DEGs upregulated under red light.

**Figure 8 ijms-23-00435-f008:**
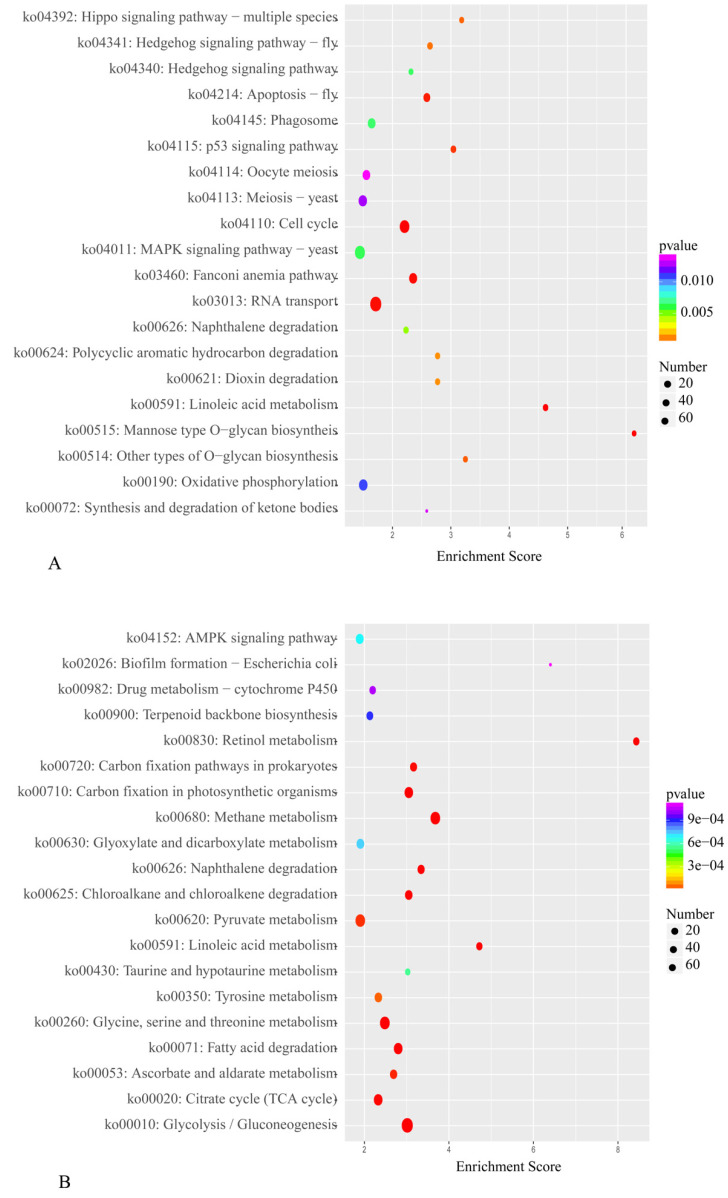
The top 20 enriched KEGG pathways from R vs. B. (**A**). Enriched KEGG pathways of downregulated DEGs. (**B**). Enriched KEGG pathways of upregulated DEGs. The color intensity is proportional to the enrichment significance, and the circle size indicates the number of enriched genes.

**Figure 9 ijms-23-00435-f009:**
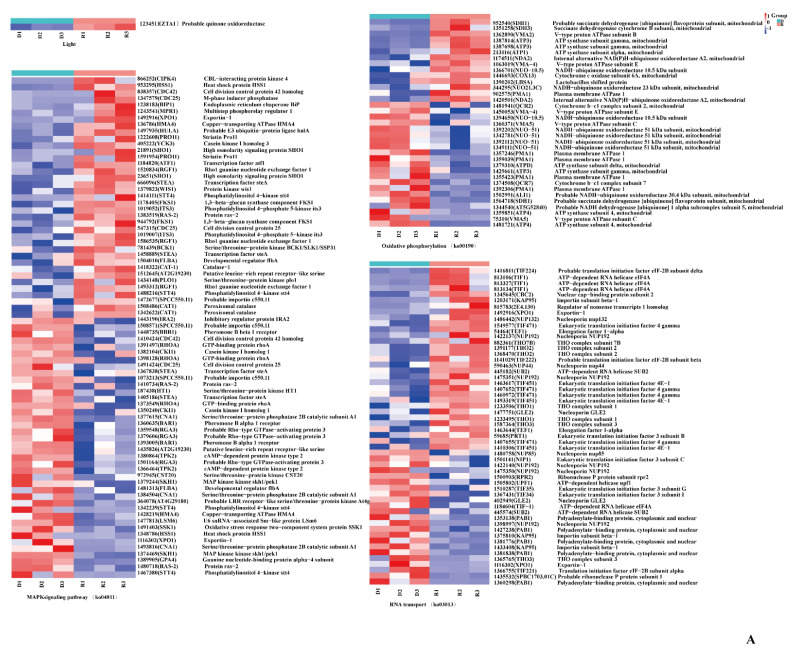
Genes involved in primordium formation in *P. eryngii*. (**A**). The DEGs between darkness and red light. (**B**). The DEGs between darkness and blue light. The gene ID number and the name of the homolog are indicated on the right side of each subplot. The gene expression values (FPKMs) were transformed to Z-score values.

**Figure 10 ijms-23-00435-f010:**
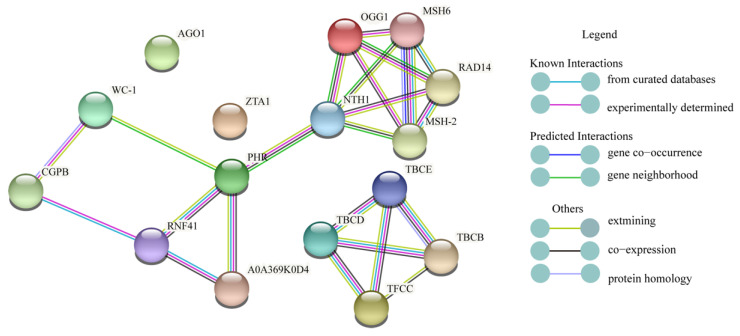
Protein-protein interaction network.

**Figure 11 ijms-23-00435-f011:**
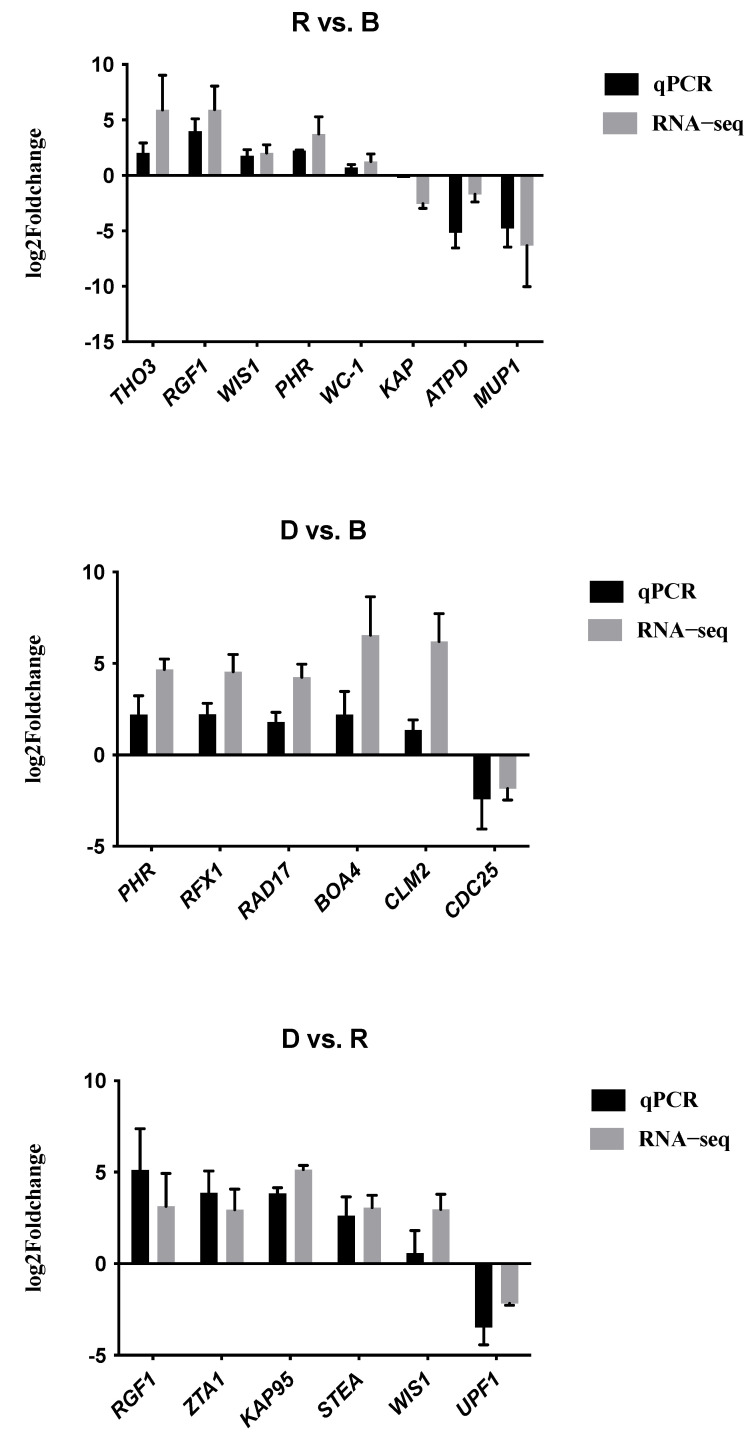
RT-qPCR validation. The X-axis shows the names of the DEGs. The left Y-axis shows the logarithm of the relative expression level. Error bars indicate the standard deviations of three biological replicates of RT-qPCR analysis.

**Figure 12 ijms-23-00435-f012:**
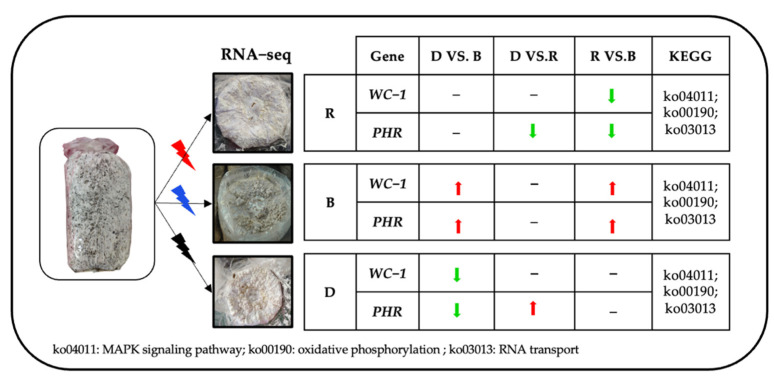
Brief summary.

**Table 1 ijms-23-00435-t001:** Summary of the transcript expression levels.

Sample	Raw Reads	Clean Reads	Valid Bases	Q30	GC	Total Mapped	Uniquely Mapped
B1	53,102,878	51,075,114	92.98%	93.61%	53.35%	46,565,408 (91.17%)	46,088,648 (90.24%)
B2	52,458,130	50,371,972	93.00%	93.59%	53.41%	45,845,104 (91.01%)	45,378,994 (90.09%)
B3	52,809,116	50,727,940	93.13%	93.54%	53.16%	46,107,974 (90.89%)	45,632,257 (89.95%)
R1	53,888,394	51,658,590	90.86%	91.57%	53.39%	47,324,146 (91.61%)	46,851,732 (90.69%)
R2	51,919,842	51,342,892	92.10%	93.07%	53.47%	47,083,554 (91.70%)	46,617,500 (90.80%)
R3	52,589,620	50,549,472	92.29%	93.47%	53.44%	45,918,518 (90.84%)	45,453,630 (89.92%)
D1	54,244,320	52,057,380	92.87%	93.37%	53.31%	47,507,321 (91.26%)	47,010,123 (90.30%)
D2	53,015,062	51,179,330	93.53%	94.05%	53.21%	46,807,561 (91.46%)	46,289,612 (90.45%)
D3	53,783,370	51,820,220	93.42%	94.38%	53.39%	47,417,325 (91.50%)	46,916,526 (90.54%)

Note: B = Blue-light primordium; R = red-light primordium; D = Dark primordium.

**Table 2 ijms-23-00435-t002:** DEGs enrich to GO terms related to light.

Gene ID	Gene Name	Foldchange	Pval	GO ID	GO Term	Description
1391851	*PHR*	135.255861	0.00109307	GO: 0003904, GO: 0018298	deoxyribodipyrimidine photolyase activity, protein-chromophore linkage	Deoxyribodipyrimidine photolyase
1391738	*PHR*	15.2406249	6.29 × 10^−9^	GO: 0003904, GO: 0018298	deoxyribodipyrimidine photolyase activity, protein-chromophore linkage	Deoxyribodipyrimidine photolyase
1447773	*PHR*	13.401109	0.01898976	GO: 0003904, GO: 0018298	deoxyribodipyrimidine photolyase activity, protein-chromophore linkage	Deoxyribodipyrimidine photolyase
1350178	*ZTA1*	12.9653923	0.04674992	GO: 0009644	response to high light intensity	Probable quinone oxidoreductase
1506953	*PHR*	8.97926851	0.03066054	GO: 0003904, GO: 0018298	deoxyribodipyrimidine photolyase activity, protein-chromophore linkage	Deoxyribodipyrimidine photolyase
1432018	*TFCC*	5.78562604	0.02793471	GO: 0032391	photoreceptor connecting cilium	Tubulin-folding cofactor C
1483280	*TFCC*	5.78562604	0.02793471	GO: 0032391	photoreceptor connecting cilium	Tubulin-folding cofactor C
1507938	*TFCC*	5.78562604	0.02793471	GO: 0032391	photoreceptor connecting cilium	Tubulin-folding cofactor C
1164413	*WC-1*	2.40057781	0.01738034	GO: 0009881, GO: 0018298, GO: 0003700	photoreceptor activity, protein-chromophore linkage, transcription factor activity, sequence-specific DNA binding	White collar 1 protein
1445782	*AGO1*	0.47726612	0.03675021	GO: 0010218	response to far red light	Protein argonaute 1
1486735	*OGG1*	0.31928656	0.04423718	GO: 0009416	response to light stimulus	N-glycosylase/DNA lyase
1557381	*SR45A*	0.21401009	0.04503058	GO: 0009644	response to high light intensity	Serine/arginine-rich splicing factor SR45a
1234511	*ZTA1*	0.02342917	0.00013587	GO: 0009644	response to high light intensity	Probable quinone oxidoreductase

**Table 3 ijms-23-00435-t003:** Primers for RT-qPCR.

GENE	Description	Primer (5′-3′)
*GAPDH*	Glyceraldehyde-phosphate dehydrogenase	F: ACGATGTCCGACGATGAG
		R: GACGGCGATGTTGGTGAA
*THO3*	THO complex subunit 3	F: ATGACGGTTCGAGAGACACC
		R: TGGCGCAAATATCGATGTAA
*PHR*	Deoxyribodipyrimidine photolyase	F: GATCTCAGGGTTGCGGATAA
		R: GGATGTGGAGTTCGGCTAAA
*KAP95*	Importin subunit *beta-1*	F: GTGTGAGGCAACCCAAAACT
		R: CCGTAGTCCTGAGCCTCTTG
*UPF1*	ATP-dependent helicase *upf1*	F: CCCCATCCAATTGTAACCTG
		R: AACCAGAATGGATGGCAGTC
*MUP1*	High-affinity methionine permease	F: CTGGTGATAGCACCCTTCTTAC
		R: CCCAGCACGATGATACCAATAC
*ATPD*	ATP synthase subunit delta	F: GTTAACATCTCGGCCGCTAC
		R: AGGGCCTCTTGAAGATTGGT
*RFX1*	Transcriptional regulator RFX1	F: CATGCGAGACTTGACCATTAGA
		R: TACTACACCGGATTGAGCTTTG
*RGF1*	Rho1 guanine nucleotide exchange factor 1	F: GCGTTTCCTGAACCACAAAT
		R: CTTCGTCGTGATCTGCGTAA
*RAD17*	Checkpoint protein rad17	F: CTGTTCACCCAAAGAAGGTAGA
		R: CGCCAGGATCTTTCGGTATT
*CDC25*	M-phase inducer phosphatase	F: CCGAAGATCCACTACCCTGA
		R: GATGCCATCGCCGTAAGTAT
*STEA*	Transcription factor *steA*	F: CGGCCAGTACGTAACATGAA
		R: ATCGTGAGGCACTGAGAACC
*WIS1*	Protein kinase wis1	F: TCCATCGAGATGTGAAACCA
		R: TATTTTGGGATTCGCCTTTG
*BOA4*	Cytochrome P450 monooxygenase BOA4	F:CTGGATAGGATGGACGAAGATTAG
		R: GACGAAGAGACGAGTTGAAGAG
*CLM2*	Cytochrome P450 monooxygenase CLM2	F: GAAGGAGGTGTTGAGATGGAAT
		R: CGATCCTTTCGGGATGAAGTAA
*ZTA1*	quinone oxidoreductase	F: CCAGCAGTCTTGGGTAAAGAA
		R: A GGTGTACTTGCGAGCTTGAT

## Data Availability

Not applicable.

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
