# Peer review of "Transcriptome Analysis Reveals Candidate Genes Involved in Light-Induced Primordium Differentiation in Pleurotus eryngii"

_ijms, 2021, doi:10.3390/ijms23010435_

Round 1

Reviewer 1 Report

In this manuscript author did the transcriptome analysis that reveals candidate genes involved in light-induced primordium differentiation in Pleurotus eryngii. In this study, primordium expression profiles under blue-light stimulation, red-light stimulation, and exposure to darkness were compared using high-throughput sequencing. A total of 16321 differentially expressed genes (DEGs) were identified from three comparisons. GO enrichment analysis showed that an amount of DEGs were related to light stimulation and amino acid biosynthesis. KEGG analyses demonstrated that the MAPK signaling pathway, oxidative phosphorylation pathway, and RNA transport were most active during primordium differentiation. Furthermore, it was predicted that the WC-1 blue-light photoreceptor and PHR play important roles in the primordium differentiation of P. eryngii. Taken together, the results of this study reveal the mechanism by which light induces primordium differentiation and provide a foundation for further research on fruiting body development in P. eryngii. Manuscript needs some English editing.

  1. The introduction is short. The author should include recent studies such as: a.Genome-Wide Identification, and Characterization of PIN-FORMED(PIN) Gene Family Reveals Role in Developmental and Various Stress Conditions in Triticum aestivum. b. Genome-wide identification and expression pattern analysis of the KCS gene family in barley. C. Genome-wide identification and characterization of abiotic stress-responsive lncRNAs in Capsicum annuum. d. Genome-Wide Identification and Characterization of the Brassinazole-resistant (BZR) Gene Family and Its Expression in the Various Developmental Stage and Stress Conditions in Wheat (Triticum aestivum). e. Genome-wide identification and functional characterization of natural antisense transcripts in Salvia miltiorrhiza. f. Genome-wide identification and expression analysis of the AT-hook Motif Nuclear Localized gene family in soybean.
  1. It would be better if the author tried to validate the function of at least one gene they found in this study.
  2. Make one hypothetical figure which depicts the findings of this study.
  3. Do in-silico protein-protein interaction study.
  4. Gene name in the manuscript should be italic. For example, in figure 10.

Reviewer 2 Report

Dou Ye and colleagues present an article on the expression of genes acting during differentiation of Pleurotus eryngii primordia, under blue/red-light stimulation and exposure to darkness. Appropriate GO-enrichment and KEGG analyses were performed in order to unravel putative pathways regulating these processes. Validation of transcriptomic data was performed by qPCR of randomly selected genes.

In general the manuscript is carefully and comprehensibly written and the overall presentation follows the typical pattern seen in articles dealing with transcriptomic analyses. Authors have previously published an article dealing with transcriptomic analyses during pileus morphogenesis under different light conditions in the same organism (DOI: 10.1016/j.ygeno.2019.09.014). In order to highlight novelty in this study, authors should at least cite, compare and discuss their results in the different developmental stages.

A scheme containing the proposed mechanism of regulation should accompany the discussion.

Results from other studies dealing with gene expression in Pleurotus eryngii under different physiological conditions (e.g. DOI: 10.3390/jof7060426 ; DOI: 10.3390/molecules21050560 ; DOI: 10.1016/j.chemosphere.2018.03.011 ; DOI: 10.1007/s00253-019-10228-z) could also be discussed.

Information on related light-dependent genes from ascomycetes could also be discussed.

Please rephrase the last sentence of the Abstract. The proposed mechanism is speculative.

Round 2

Reviewer 1 Report

There are improvements in the manuscript, however, still, a lot can be improved.

  1. The author did changes I asked however overall manuscript English still needs to be improved to meet journal publishing standards.
  2. Overall figure quality is very poor and small. It is very difficult to read them. Please enlarge them or divide them into two.
  3. Why there are no error bars on qRT-PCR data. Did the authors use a triplicate sample? If yes then why not write in the legend.
  4. Protein-Protein interaction meant only using a webserver like strings to predict interacting partners, not doing real-time experiments.

Reviewer 2 Report

The manuscript has certainly improved, however, the major concern of this reviewer is still not addressed. In this context, as also pointed out in the first report, authors have previously published an article with transcriptomic analyses in the same fungus, in similar conditions and in the developmental stage of pileus morphogenesis. Are the genes affected during the stage of primordium differentiation the same as the ones affected during pileus morphogenesis? Are there no major differences? Are the proposed pathways equally involved in the response to light? In my opinion, there is no point in presenting this manuscript without a more detailed comparison between the two transcriptomes.

Round 3

Reviewer 1 Report

I am happy with the author's comments. The manuscript looks refined now and can be accepted in its current format.

Reviewer 2 Report

The manuscript has improved significantly. I have no further comments for the current article. Unfortunately, no differences between the RNA seq data in the different developmental stages could be highlighted herein. On the other hand, authors are lucky to possess expression data from similar conditions in two different developmental stages of the same organism. If authors are interested in approaching the mechanism of light induction during development, then my advice for eventual future studies would be to thoroughly compare their data sets, analyze in depth and try to identify even minor changes in the expression patterns of related or even unrelated genes.